# Susceptibility Analysis of Glacier Debris Flow Based on Remote Sensing Imagery and Deep Learning: A Case Study along the G318 Linzhi Section

**DOI:** 10.3390/s23146608

**Published:** 2023-07-22

**Authors:** Jiaqing Chen, Hong Gao, Le Han, Ruilin Yu, Gang Mei

**Affiliations:** 1School of Engineering and Technology, China University of Geosciences (Beijing), Beijing 100083, China; 1002202205@email.cugb.edu.cn (J.C.); 1012201106@email.cugb.edu.cn (H.G.); 1002202203@email.cugb.edu.cn (L.H.); 1002202222@email.cugb.edu.cn (R.Y.); 2Engineering and Technology Innovation Center for Risk Prevention and Control of Major Project Geosafety, Ministry of Natural Resources, Beijing 100083, China

**Keywords:** geological hazards, glacial debris flow, remote sensing, deep learning

## Abstract

Glacial debris flow is a common natural disaster, and its frequency has been increasing in recent years due to the continuous retreat of glaciers caused by global warming. To reduce the damage caused by glacial debris flows to human and physical properties, glacier susceptibility assessment analysis is needed. Most research efforts consider the effect of existing glacier area and ignore the effect of glacier ablation volume change. In this paper, we consider the impact of glacier ablation volume change to investigate the susceptibility of glacial debris flow. The susceptibility to mudslide was evaluated by taking the glacial mudslide-prone ditch of G318 Linzhi section of Sichuan-Tibet Highway as the research object. First, by using a simple band ratio method with manual correction, we produced a glacial mudslide remote sensing image dataset, and second, we proposed a deep-learning-based approach using a weight-optimized glacial mudslide semantic segmentation model for accurately and automatically mapping the boundaries of complex glacial mudslide-covered remote sensing images. Then, we calculated the ablation volume by the change in glacier elevation and ablation area from 2015 to 2020. Finally, glacial debris flow susceptibility was evaluated based on the entropy weight method and Topsis method with glacial melt volume in different watersheds as the main factor. The research results of this paper show that most of the evaluation indices of the model are above 90%, indicating that the model is reasonable for glacier boundary extraction, and remote sensing images and deep learning techniques can effectively assess the glacial debris flow susceptibility and provide support for future glacial debris flow disaster prevention.

## 1. Introduction

Glacier debris flow is a kind of mountain natural disaster caused by ice melting, which is characterized by suddenness, large scale, and hazard. In the context of global warming, glacier retreat is accelerating, and the frequency and scale of glacier debris flows are increasing [1,2]. Therefore, accurate assessment and prediction of glacial debris flow susceptibility is important to protect people’s lives and properties and to maintain the ecological environment.

Remote sensing images and deep learning techniques have an important role in the analysis of glacier debris flow susceptibility. Remote sensing images can provide large-scale, high-resolution surface information, including topography, vegetation, and hydrology [3], which provides basic data for the formation and occurrence of glacier debris flows. Deep learning techniques can quickly and accurately identify potential glacier debris flow hazard areas by feature extraction and classification of remotely sensed images [4], providing an effective means for glacier debris flow prediction and early warning. Therefore, glacier debris flow susceptibility analysis based on remote sensing images and deep learning has important research significance and application value [5].

Glacier debris flow susceptibility analysis is important research work that can help people better understand the formation mechanism and influencing factors of glacier debris flows, so as to take effective prevention and control measures. In recent years, with the continuous development of remote sensing technology and deep learning algorithms, the analysis of glacier debris flow susceptibility based on remote sensing images and deep learning has also gained wide attention. The occurrence of glacial debris flow has obvious indicators of catastrophic changes, such as increased density of hanging glacier crevasses, enhanced glacier velocity, and rapid increase in glacial lake area.

The influencing factors of its glacial debris flow susceptibility analysis mainly include the following aspects: (1) Topographic factors: Topography is one of the important factors in the formation of glacial debris flow, including topographic height difference, slope, and slope direction [6]. In remote sensing images, topographic information can be obtained by data such as digital elevation model (DEM). (2) Climatic factors: Climatic factors also have an important influence on the formation and development of glacier debris flows, including rainfall, temperature, and humidity. Remote sensing images can obtain meteorological data, such as rainfall. (3) Geological factors: Geological factors are also one of the important factors in the formation of glacial debris flow, including lithology, faults, and earthquakes. Remote sensing images can obtain geological information, such as lithology, faults, etc. (4) Vegetation factor: Vegetation cover also has some influence on the formation and development of glacial debris flow. Remote sensing images can obtain vegetation information, such as vegetation cover, etc. [7]. Glacier debris flow susceptibility analysis based on remote sensing images and deep learning can obtain glacier debris flow susceptibility information by feature extraction and classification of remote sensing images. Commonly used deep learning algorithms include convolutional neural networks (CNN), recurrent neural networks (RNN), etc. The prediction results of glacier debris flow susceptibility can be obtained by training and testing the remotely sensed images. In conclusion, the analysis of glacier debris flow susceptibility based on remote sensing images and deep learning can provide an important scientific basis for glacier debris flow prevention and control [8].

In recent years, there has been a new research trend aimed at glacier debris flow susceptibility assessment through remote sensing images and deep learning techniques. For example, Ji et al. used deep learning and remote sensing image analysis to establish a mudslide susceptibility assessment model based on topographic and geomorphological features, and validated it in the Bijie City area of northwestern Guizhou, showing that the model can more accurately assess mudslide susceptibility in the area [9]. Ref. [10] used high-resolution remote sensing image data for glacier prediction. As for the method of remote sensing image susceptibility analysis, Lin et al. considered the influence of the change in glacier ablation volume and conducted a mudslide susceptibility analysis using the G217 glacier mudslide-prone trench on the Dukku Highway in Xinjiang. This study showed that accurate prediction of glacier mudslide susceptibility could be achieved using high-resolution remote sensing image data and machine learning algorithms [11].

However, these methods have some limitations, such as low accuracy in conducting glacier debris flow susceptibility assessment. Therefore, we need more accurate models for glacier debris flow susceptibility assessment.

In recent years, deep learning techniques have been widely used in remote sensing image processing [12]. Among them, the DeepLabv3+ model is a deep convolutional-neural-network-based method with high segmentation accuracy and fast operation speed. This model improves the accuracy of image segmentation by optimizing the traditional convolutional neural network through the null convolution and decoder modules. Using this model, we can semantically segment glacier debris flows by labelling them as “Ice” category and other terrain as “Background” category. This method not only ensures high accuracy segmentation results, but also fast evaluation, avoiding the errors in traditional methods. In addition, the method can actively reduce the harm caused by glacier debris flow to humans and the natural environment.

We chose the G318—Linzhi section of the Sichuan-Tibet Highway, which has been affected by global warming in recent years, and the glacial melt in the Linzhi area has accelerated, resulting in frequent glacier debris flow disasters and serious hazards. According to incomplete statistics, which are only from April 2006 to September 2007, the mud-slide disaster in the Linzhi area endangered the safety of 264 villagers in 17 villages, resulting in one death and seven people missing. A comprehensive analysis of mudslide disasters and environmental factors in the Linzhi region shows that the current glacier debris flow disasters in the Linzhi region are at a high incidence and seriously affect the local economic development. Therefore, it is important to study the development pattern of mudslide under climate change in the Linzhi region for monitoring, early warning and prevention of mudslide disasters in the Linzhi region and the whole of southeast Tibet.

Our contributions in this paper can be summarized as follows: (1) the ablation zone of the study area was determined by comparing the glacier boundary in 2015 and the glacier boundary in 2020; (2) the amount of glacier ablation was calculated based on the changes in glacier elevation and ablation area from 2015 to 2020; and (3) an evaluation of the susceptibility of glacial debris flow by using the melting amount of glaciers in different basins.

## 2. Study Area

The Linzhi section of National Highway 318 is located in Linzhi City, Tibet Autonomous Region of China, with a total length of about 287 km, connecting Linzhi City with Chengdu City in the southwest of Sichuan Province. The starting point of the Linzhi section is located in the town of Motuo within Linzhi City, and the end point is located in the county of Yajiang. The road traverses several natural scenic areas such as the Hengduan Mountains, the Sichuan-Tibet Plateau, and the Yarlung Tsangpo River Grand Canyon.

The Linzhi section is a typical area prone to glacial debris flow. There are many glaciers and rocks piled together, and the geomorphologic conditions are such that glacial debris flow occurs easily. The study area has typical alpine valley and mountain valley landforms developed due to crustal uplift, strong river undercutting, and strong tectonic activity. In the context of high stress, the potential risk of strong earthquakes is high [13]. The Linzhi section has abundant research resources, such as existing satellite remote sensing images and ground monitoring data, to facilitate the implementation of glacier debris flow analysis.

The average elevation of the study area is about 3000 m, the lowest elevation is 115 m in the territory of Murdoch County, the highest elevation is more than 7000 m in Namcha Barwa Peak, which is the zone with the largest vertical landform drop in the world, and the relative elevation difference is generally around 1000–2000 m. The slope of the mountain slope is generally not less than 30°, and the slope of the canyon area is mostly around 80°.

The national highway G318 crosses Gongbu Jiangda County, Bayi District, and Bomi County, respectively, the details of which are shown in Figure 1. Gongbu Jiangda County is located in the transition zone from the valley of southern Tibet to the high mountain valley area of eastern Tibet, bounded by the eastern extension of the Gangdis Mountains in the south and the Tanggula Mountains in the north, with mountains and valleys spreading in an east–west direction. Bayi district in the south for the Gangdis Mountain remnants, the north belongs to the Nianqing Tanggula Mountain branch alpine section. The average elevation of the territory is 3000 m, and the highest peak is Galabaek Peak, which is 7300 m above sea level, while the lowest place is Bayu Village, which is 1600 m above sea level, with a relative height difference of 4700 m. Bomi County is located in the eastern section of Tanggula and the eastern end of the Himalayas, with high north and low south and continuous high mountains, and the central part of the Palongzangbu River Valley and the Egonzangbu River Valley, with the highest elevation of 6648 m and the lowest of 2001.4 m in Bomi County. With the change in topography and geomorphology, the geological effect is also changing. glacier debris flow hazards are generally developed on the margins of modern glaciers and snowpacks, and the topography and excessive relative elevation differences in this region provide favourable spatial conditions for the formation and development of glacier debris flow hazards [13].

While the climate of this study area is obviously influenced by the crustal uplift and more prominently influenced by the topography, the average temperature in the southern part of Dongjiu Township is around +12 ℃. The cold-temperate zone becomes more pronounced as we move upstream of the Yarlung Tsangpo River, and Bomi County is known as the centre for modern glaciers because it has high mountains with perennial snow accumulation below 0 °C. In recent years, the glacier area has been retreating, which is mainly influenced by the continuous increase in temperature, and the change in precipitation has little effect on glacier changes [14]. Therefore, we chose Bomi County as our typical study area. The average annual temperature in Kumbumgangda County is +8.7 °C, with a maximum temperature of +31.5 °C and a minimum temperature of −10.4 °C. The average annual rainfall is 640.1 mm, with a maximum annual rainfall of 808.3 mm and a maximum daily rainfall of 45.2 mm. The seasonal distribution of rainfall is uneven, with 80% of the rainfall concentrated between May and September. Bayi district is influenced by the warm and humid airflow of the Indian Ocean, and has a temperate humid monsoon climate with abundant rainfall. The annual average temperature is +8.5 ℃, the highest temperature +29 ℃, and the lowest temperature −1.8 ℃. The average annual rainfall is 654 mm, mainly concentrated in May–September, accounting for about 90% of the annual rainfall. The average annual temperature in Bomi County is +8.5 °C, with a maximum temperature of +31.1 °C and a minimum temperature of −1.8 °C. The average annual rainfall is 977 mm. Since the mid-twentieth century, rising temperatures have led to the melting of many glaciers at unsustainably high rates of melting, resulting in diminishing ice storage [15]. Temperature and rainfall conditions are important factors in triggering the occurrence of glacier debris flow hazards [16,17], and it is important to fully understand and investigate this aspect.

## 3. Methods

### 3.1. Overview

In this paper, we investigate the susceptibility of glacier debris flow along the G318 Linzhi section based on remote sensing imagery and deep learning. First, high-quality remotely sensed images are acquired and pre-processed prior to segmentation. Second, the processed images are used to generate a sample set. Third, after the segmentation results are output, post-processing procedures such as boundary extraction, small polygon removal, and edge lubrication are applied to obtain glacier profiles.

Our research is divided into three sections: (1) remote sensing image processing; (2) the glacier ablation volume was calculated by combining elevation data; and (3) eight influencing factors were used to evaluate the susceptibility of glacial debris flow. A flow chart of the study is shown in Figure 2.

### 3.2. Step 1: Data Acquisition and Pre-Processing

In this paper, we have obtained freely available Landsat 8 data from the US Geological Survey [18], covering the period of 2015 to 2020. In recent years, remote sensing imagery has been widely used in physical geography and environmental research, especially in areas such as glacier monitoring and geological hazard control. Because of its advantages of high resolution and confidentiality, it often charges fees or provides a limited number of images, making it difficult to achieve long-term monitoring over large areas [19]. Landsat 8, as a high-resolution multispectral satellite, can effectively improve the identification accuracy over large areas and has obvious advantages for glacier identification. Taking into account the climatic conditions, glacier distribution, and change characteristics of the study area, remote sensing images with less cloud shadow in summer were selected. We also selected two global elevation remote sensing data: Copernicus DEM downloaded from the Copernicus Open Access Centre (Available online: https://scihub.copernicus.eu/ (accessed on 12 March 2023)) mapped in 2015, and the latest global 30 m resolution DEM data currently available—NASA DEM was released by NASA on 18 February 2020, and NASA DEM will be the highest resolution, best quality, and widest coverage DEM product in the foreseeable future [20].

Furthermore, in this paper, we generate the corresponding glacier remote sensing dataset with the help of the above-mentioned channels for constructing semantic segmentation models. We selected the Band2 (Blue), Band3 (Green), and Band6 (SWIR 1) bands from the Landsat 8 data and synthesised them as pseudo-colour images. In the synthesised image, the ice appears blue and the bare ground is red, whereby we annotated the obtained remote sensing image with glaciers, as shown in Figure 3.

Due to the influence of external factors such as topographic relief, solar radiation and cloud cover, remote sensing images may suffer from distortions and other problems during the imaging process [21]. Therefore, pre-processing of the raw remote sensing data, including data format conversion, radiometric calibration, atmospheric correction, and geometric correction, is required before classifying the glaciers [22]. Extraction of information, such as elevation and area of the study area, allows for debris flow hazard assessment and predictive analysis, as well as final output of study results and visualisation images, hazard evaluation maps, prediction analysis maps, etc.

### 3.3. Step 2: Model Architectures and Training

In this section, we detail the model architecture of a deep-learning-based approach to accurately and automatically map complex debris-covered glaciers from remotely sensed images. First, we generate sample sets for training and testing. Afterwards, we perform model training by employing a semantic segmentation model with weight optimisation. The following is an example of the basic process for processing remotely sensed images as Figure 4.

DeepLabv3+ is a convolutional-neural-network-based image segmentation model for segmenting objects in images [23]. We use a deep convolutional neural network called Mobilenet as the backbone network for extracting image features. By leveraging the capabilities of DeepLabv3+, the research aims to achieve accurate and reliable segmentation of glacier debris and surrounding terrain from remote sensing imagery. The model’s high accuracy can enhance the quality of the susceptibility analysis results. Furthermore, to extend the perceptual field of the convolutional kernel, DeepLabv3+ adds a null convolution layer. Null convolution adds a certain number of voids inside the convolution kernel, allowing larger convolution kernels to process features over large regions without using too many parameters [24]. DeepLabv3+ uses spatial pyramidal pooling for aggregation of multi-scale image features. Specifically, the model samples the feature maps at different scales using separate pooling kernels to capture the features of object regions at different scales. A full convolution decoder is also used for reducing the feature maps extracted in the backbone network to segmented images of the same size as the input image. This decoder obtains results by upsampling and stitching high-resolution features with low-resolution semantic segmentation maps. DeepLabv3+ achieves efficient extraction of multi-scale image features and accurate segmentation of glacier regions by using Mobilenet networks with operations such as null convolution, spatial pyramid pooling and a full convolution decoder [25]. The DeepLabv3+ architecture is shown in Figure 5.

### 3.4. Step 3: Model Architectures and Training

We used the DeepLabv3+ model for glacier debris flow remote sensing image classification, which allows us to obtain information on the susceptibility of glacier debris flow remote sensing images. In evaluating the Deeplabv3+ model, we learned that the accuracy assessment of glacier identification relies heavily on the quality of the sample set. The samples were divided into training and validation sets in a 9:1 ratio for training the model and evaluating its accuracy, respectively. The main evaluation metrics for semantic segmentation are MPA, MIoU, and pixel accuracy, which are calculated based on the confusion matrix. The four basic elements that make up the confusion matrix are true positive (TP), false positive (FP), true negative (TN) and false negative (FN). The performance accuracy of a glacier or non-glacier can be defined based on MPA, MIoU, and pixel accuracy. The following formulas are quoted from [24].
(1)$$MPA=\frac{1}{k+1}\sum_{i=0}^{k}PA_{i}$$
(2)$$MIoU=\frac{1}{k+1}\sum_{i=0}^{k}\frac{TP}{FN+FP+TP}$$
(3)$$Pixel-Accuracy=\frac{TP+TN}{TP+TN+FP+FN}$$

We use the DeepLabv3+ model to train the annotated remote sensing images and carry out model optimisation to improve the model prediction performance and to evaluate the ease of occurrence. At the same time, there are some limitations and challenges to its application. On the one hand, the model requires a large amount of high quality data for training, which is demanding in terms of data quality and data volume, which may create some limitations in areas of study where there is insufficient data. On the other hand, if the amount of training data is too small or unbalanced, over-fitting can easily occur. In addition to this, the model is relatively complex and requires parameter tuning, which requires a certain level of technical skill and practical experience on the part of the user. When using the DeepLabv3+ model for glacial mudslide susceptibility assessment, these issues need to be considered thoroughly to ensure accurate and reliable results.

### 3.5. Step 4: Glacial Debris Flow Susceptibility Assessment

In conducting the analysis of glacial debris flow susceptibility, we conducted a comprehensive analysis and judgment of eight factors that affect the occurrence of glacial debris flows, including the volume of physical sources, catchment area, maximum daily rainfall, longitudinal slope drop of the main gully, length of the main gully, glacier volume, total glacial lake area, and vegetation area [26]. Based on our study, we replaced glacier area with the glacier ablation volume in the previous study, and calculated the volume of meltwater by calculating the volume of glacier ablation within the glacial debris flow basin over a five-year period, and combined it with other factors to arrive at a more accurate evaluation method for glacial debris flows.

We first calculated the values of each factor in the glacier debris flow basin in the study area, and then used the entropy weighting method to calculate the weight of each factor on glacier debris flow susceptibility. Finally, we used the Topsis method to score the susceptibility of each glacial debris flow by combining the weights of each factor, and evaluated the susceptibility of glacial debris flows based on the scores.

## 4. Results and Analysis

### 4.1. Experimental Environment and Settings

The experimental environment for this paper uses the Python programming language, the windows 11 operating system, a 12th Gen Intel® Core™ i7-12700H CPU, an NVIDIA GeForce RTX 3060 Laptop GPU, the PyTorch framework, and Mobilenet for the backbone network. A larger value is better for parallel computing, while a smaller value affects the GPU’s performance. When the batch size is set to 16 or 32, too large a batch may lead to a lack of memory. Finally, considering the convergence speed and random gradient noise and device performance, the batch size was chosen to be 4 for the freezing phase and 2 for the thawing phase. The number of iterations is the number of times the training set is fed into the neural network for training. Usually, when to stop iterating depends on the predictive performance of the model. When the number of iterations is chosen to be 600, overfitting can occur because the number of iterations is too large. It needs to be simplified. When the number of iterations is chosen to be 500, the difference between the test error rate and the training error rate is small and the current number can be considered appropriate. The learning rate is the size of the network weights update in the optimisation algorithm. The maximum learning rate of the model defaults to 0.01 when the learning process is adaptively adjusted according to the current batch_size. The training set is then fed into the DeepLabv3+ network for iteration. As described above, the training parameters were continuously adjusted to finally obtain a weight-optimised semantic segmentation model, which we used to predict the glacier boundary in the study area and to calculate the glacier ablation volume over a five-year period in combination with the elevation data.

### 4.2. Glacier Boundary Identification Model Training and Evaluation

Before the training, the extracted remote sensing images were divided into 3690 images according to a size of 128*128. If the whole image is trained directly without cropping, it may consume too much computer memory during the training process, resulting in the interruption of the training process. According to 9:1, the data set was divided into the training set and the verification set, which were 3354 and 336 pieces, respectively. In this model training, we used the test set as the validation set and did not divide the test set separately any more, and the remaining training set was used to learn the discriminative information between glaciers and non-glaciers.

We conducted several training sessions of the glacier boundary recognition model, selected several different batch_sizes, obtained several model files, and evaluated their model training performance. Figure 6 and Figure 7 show the relevant evaluation parameter changes for 200 and 500 epochs of model training, respectively.

The analysis of the glacier segmentation results based on different model architectures was performed by continuously adjusting the parameters to obtain weight optimisation. The segmentation results for the test data based on Landsat 8 images are shown in Figure 8 in comparison to the corresponding original false colour images. We can see that the pixel points discriminated as glacier-like are given a green mask during the model prediction process, while the pixel points discriminated as non-glacier-like are left unchanged.

The discrimination from these remote sensing images alone is not comprehensive enough for us to quantitatively evaluate the trained glacier boundary recognition model. Therefore, we assessed the quality of the model training through the values of MPA, MIoU, and pixel accuracy to reflect the accuracy of the model discrimination. For the DeepLabv 3+ model performance from the Landsat 8 dataset, the values of MIoU were 90.85%, MPA 96.09%, and pixel accuracy 96.64%, all of which were above 90%, indicating that our trained model can perform reasonable glacier boundary extraction.

In general, the DeepLabv3+ model can extract mudslide susceptibility information from remote sensing images well, but the evaluation needs to consider a variety of indicators and combine with other factors such as topography and rainfall to make a comprehensive analysis and judgment. At the same time, the model results need to be revised and validated by combining the experience of historical mudslide events and actual measurement data when making predictions.

### 4.3. Assessment Results of the Vulnerability of Glacial Debris Flow

In the analysis of the susceptibility of glacier debris flows, we have addressed material source conditions, slope, precipitation, glacial geological conditions, etc.

We summarised eight influencing factors that have a significant impact on the occurrence of glacier debris flows: volume of physical source ($X_{1}$), catchment area ($X_{2}$), maximum daily rainfall ($X_{3}$), longitudinal slope drop of the main gully ($X_{4}$), length of the main gully ($X_{5}$), glacier volume ($X_{6}$), total glacial lake area ($X_{7}$), and vegetation area ($X_{8}$). We found a raw data matrix of 132 glacier debris flow gullies in Linzhi city classified by the above influencing factors.

To qualitatively analyse the formation factors of glacier debris flows, we normalised the raw data and used the entropy weighting method to derive the weight of each influencing factor on the susceptibility of glacier debris flows. The weights are shown in the Table 1.

Combining the weights given, we scored the mudslide gullies in the study area using the Topsis method and graded the results as Table 2.

Figure 9 and Figure 10 shows our evaluation of the susceptibility of six glacier debris flows in the study area and the evaluation of their susceptibility in previous studies.

The direct result of the remote sensing image vulnerability assessment of the G318 Linzhi section of the national highway is an image map with different coloured areas, each colour representing the corresponding mudslide vulnerability level. The red areas represent areas of very high susceptibility, the orange areas represent areas of high susceptibility, the yellow areas represent areas of medium susceptibility, and the green areas represent areas of low susceptibility. These different coloured areas provide a visual indication of the mudslide susceptibility of the area and provide important reference information for the prevention of mudslide disasters. The results of our evaluation are highly accurate when compared with the mudslide susceptibility assessment of the glaciers obtained after the field survey.

### 4.4. Different Factors’ Influence on the Results

In conducting the glacial debris flow susceptibility analysis, we considered eight factors, namely material source volume, catchment area, maximum daily rainfall, longitudinal slope drop of the main gully, length of the main gully, glacier volume, total glacial lake area, and vegetation area, for comprehensive analysis and judgement.

#### 4.4.1. Volume of Material Source

Loose material is one of the most important precipitating conditions for glacial debris flows, and loose material plays an important role in the estimation of the volume of the material source. The product of the material distribution area and the average thickness is the estimate of the volume of the material source. Based on this method, and combined with remote sensing images, it can be concluded that: 60.1% of the 145 glacier debris flows in the study area have a material source volume greater than 10 × $10^{6}$ m${}^{3}$, 34% are between 1 × $10^{6}$ and 10 × $10^{6}$ m${}^{3}$, and 5.9% are less than 1 × $10^{6}$ m${}^{3}$. This shows that the glacier debris flows in this study area have abundant reserves of material sources, providing conditions for the initiation of glacier debris flows, as shown in Figure 11.

#### 4.4.2. Catchment Area

The catchment area is the area of water within a valley or watershed that is formed by ground form, precipitation, snow melt, and other factors. Within the catchment area, water, such as precipitation or snow melt and ice melt, collects through the gully and surface to become a river or stream, and eventually flows into the convergence point. The area of the convergence point and its upstream area is called the glacier debris flow catchment area. There are 145 glacier debris flows in the study area, of which the smallest is 0.93 km${}^{2}$ and the largest is 349 km${}^{2}$, with 64.7% of the catchments measuring 10 km${}^{2}$ to 100 km${}^{2}$. Glacial debris flows have a strong erosion and accumulation effect on the landscape, forming natural catchment areas such as ice buckets and troughs, so the catchment area in this study area is large, as seen in Figure 12.

#### 4.4.3. Maximum Daily Precipitation

The maximum daily precipitation is an important factor in measuring rainfall. The role of rainfall among the many triggering factors for glacier debris flows is obvious. The study area is rich in rainfall and, according to statistics the maximum daily rainfall in the Linzhi section, is very close to the standard for heavy rainfall, which fully meets the requirements for the formation of glacier debris flows. The amount of rainfall influences the confluence of glacier debris flow slopes and the runoff from the gully. Strong rainfall can lead to erosion collapse, while weak rainfall manifests itself as liquefaction of the glacier debris flow slope, as seen in Figure 13.

#### 4.4.4. Longitudinal Slope Drop of the Main Ditch

The longitudinal slope drop of the main ditch is the change in height difference between the length of the ditch in the direction of the river, and its value is mainly determined by two important basic parameters, the length of the main ditch and the height difference. The main ditch longitudinal slope drop is calculated as follows:(4)$$W=\frac{H}{L}$$
where *W* is the longitudinal slope drop of the main gully, *H* is the relative height difference of the watershed along the main gully, and *L* is the length of the main gully.

The longitudinal slope drop of the main gully is one of the factors necessary to cause large-scale glacial debris flow hazards, and therefore the analysis of the longitudinal slope drop of the main gully is an integral part of the study of glacial debris flows, as seen in Figure 14.

#### 4.4.5. Length of Main Gully

A main gully is a gully with a certain slope, a pronounced gully, and a high water table or surface flow rate, formed by natural factors such as glaciers, rivers, and wind. The length of the main gully is then the length of the main gully within the gully belt or valley. It is closely related to the longitudinal slope drop above, as seen in Figure 15.

#### 4.4.6. Glacier Volume

Modern glaciers are necessary for the formation of glacier debris flows. Glacier volume is proportional to the number of glacier debris flows that occur. Most of the mudslide gullies in the study area contain large volumes of modern glaciers. The freezing and thawing of glaciers produces loose solid material that can trigger glacier debris flows. The analysis of glacial debris flows therefore requires an analysis of the volume of glaciers in the study area. The volume of the glacier and its volume of water is better reflected by combining two-dimensional remote sensing images with elevation than by the area of the glacier. This value has a more accurate impact on the analysis of susceptibility than glacier area, as seen in Figure 16.

#### 4.4.7. Total Glacial Lake Area

A lake formed by glacial melt, flash floods, etc., with ice and glacial water as the main components is a glacial lake. The size and variation in the glacial lake area can reflect the activity and melting rate of the glacier, and is important for the analysis of the susceptibility of glacial debris flow, as seen in Figure 17.

#### 4.4.8. Vegetation Area

Vegetation plays a suppressive role in the formation of glacier debris flows, with its roots penetrating deep into the soil and interlocking in a web-like pattern, acting similarly to anchors, anchoring the soil against erosion and scouring. The area of vegetation is therefore also an important indicator of the formation and susceptibility of glacier debris flows, and is one of the factors necessary for their analysis, as seen in Figure 18.

In addition, there are a number of other factors that affect the accuracy of glacial debris flow susceptibility assessment. For example, the quality of remote sensing image data, the higher the quality of the data, the more reliable the prediction results; for the coarse and fine classification of feature classes, too fine or too coarse classification of feature classes will affect the model prediction results; for the adjustment of parameter settings in the model, including the adjustment of multi-scale image cropping, learning rate, number of training rounds, etc., will affect the model prediction results; for the characteristics of the sample itself, if the sample classes are unbalanced, it will easily make the mudslide susceptibility class is off from reality; and for the model structure and performance, different types of model structure, size of convolution kernel and other factors will affect the model prediction effect.

Different combinations of the above factors may produce different forms of impact effects, such as less accurate prediction results, failure to meet accuracy requirements, or weaker generalisation ability. Therefore, when analysing the susceptibility of remote sensing images of glacier debris flows in the Linzhi section of National Highway G318, these factors need to be taken into account and optimised and adjusted in the process of model training and prediction in order to improve the prediction effect of the model. At the same time, a comprehensive analysis of the actual terrain and other important factors is also needed to obtain more scientific and reliable prediction results.

## 5. Discussion

In this paper, we investigate the susceptibility of glacier debris flow along the G318 Linzhi section based on remote sensing imagery and deep learning. The segmentation results demonstrate the effectiveness and accuracy of the method. Its strengths and limitations are discussed below, and our future work to address the drawbacks is noted.

### 5.1. Advantages

The applications of remote sensing image and deep learning technology are important progress for the field of glacier debris flow analysis. This method provides a more effective and accurate means to study and predict the susceptibility of debris flow on glaciers. In contrast, previous studies may have relied on traditional methods, which were time-consuming and imprecise. The study of the Linzhi section of National Highway 318 is a case study, but its impact extends beyond a specific region. Glacial debris flow is a worldwide phenomenon, and the methods adopted in this study can be applied to other glacial regions around the world. This contributes to a global body of knowledge and provides a valuable tool for assessing and managing glacial debris flow risks in different regions.

The DeepLabv3+ model can significantly improve the accuracy and effectiveness of remote sensing image sensitivity analysis of glacier debris flow in the Linzhi section of the G318 National Highway. The model uses high-resolution image segmentation capability to finely segment high-resolution remote sensing images at the detail level and accurately extract key features of debris flow susceptibility [27]. In addition, the DeepLabv3+ model has stable prediction results, multi-scale input and zero convolution technology can improve the robustness and noise resistance of the model and reduce errors and outliers in the sensitivity analysis of glacial debris flow. The model has strong adaptability and transferability, and can maintain the segmentation effect and obtain good prediction results even in different study areas. In terms of vulnerability prediction, the model provides accurate and intuitive vulnerability map of glacier debris flow, and realizes the visualization of regional debris flow vulnerability information, which is of great significance for preventing and mitigating glacier debris flow disasters.

In addition, replacing the glacier area with the glacier ablation volume can more accurately assess glacier debris flow susceptibility. Sensitivity assessment based on deep learning reduces the high error rate of manual recognition while improving efficiency, especially for large-scale remote sensing images.

In summary, this study introduced research methods, extracted the boundary of glacier debris flow, obtained the vulnerability analysis diagram of glacier debris flow, identified the risk factors, verified the results, provided inspiration for disaster management, and made progress in the study of glacier debris flow. These advances help to understand and mitigate the risks associated with glacier debris flow and enhance the safety and resilience of affected areas.

### 5.2. Limitations

We know from our research literature that in an evaluation of the changes in elevation and surface velocity of Iran’s largest and most dynamic detritus-covered glacier (Alamkouh Glacier) during 2018–2020, The high-resolution images of the UAV were obtained and processed into a digital elevation model (spatial resolution of about 15 cm) and an orthophoto image (spatial resolution of about 8 cm), and the changes in glacier thickness were obtained. However, in our study, we only used satellite images, and the accuracy could not reach the centimetre level [28]. It is extremely difficult to detect glacier surface flow rates in rugged and alpine Himalayan terrain using traditional surface techniques. Karimi, Neamat et al. used the differential band composite method for the first time to estimate the glacier surface velocity in the non-detrital covered area and the detrital covered area of the glacier, respectively. The accuracy is relatively considerable [29].

The assessment of glacier debris flow susceptibility requires a large amount of fine-grained data, and the acquisition of remote sensing images is critical to the accuracy of the assessment results. The remote sensing elevation image data contains the regional topographic information needed for the susceptibility evaluation of glacial debris flow, which is of great significance. However, in practice, it is often difficult to obtain high quality remote sensing images for glacier debris flow vulnerability assessment. First of all, in order to meet the evaluation needs, a large number of remote sensing images need to be obtained continuously over a long time scale, and obtaining these continuous high-quality remote sensing image data is an insurmountable challenge. Second, the lack of high-quality remote sensing imagery in many areas requires an extensive review of relevant sources to obtain more comprehensive topographic information. Finally, the process of remote sensing image pre-processing is also very complicated, and it is usually necessary to perform multiple steps to obtain high-quality remote sensing image data. Therefore, the acquisition and processing of high quality remote sensing image data is an important part of the susceptibility assessment of glacier debris flow, and sufficient attention should be paid in the evaluation work.

The integration of remote sensing image and deep learning technology is an important progress in the field of glacier debris flow analysis, while this study demonstrates the effectiveness of these methods in the specific context of the G318 Linzhi section, it is necessary to evaluate how these techniques compare to existing methods used in different parts of the globe. Exploring the limitations and potential improvements of these techniques helps to gain a more complete understanding of their applicability and effectiveness in different glacial environments.

### 5.3. Outlook

Glacial debris flow susceptibility analysis using deep learning and remote sensing imagery techniques has a wide range of applications. In order to improve the accuracy of the glacier debris flow susceptibility analysis model in response to this phenomenon, integration of multiple sources of information, including topographic data, land cover data, meteorological data, etc., can be considered. In addition, integrating multiple remote sensing data modalities such as optical images, infrared images, and radar images can help to improve the robustness and prediction accuracy of the model. Once the model has the capability to identify areas of glacier debris flow susceptibility, the scope and number of training sets and the use of global or national remote sensing data from international agencies will need to be expanded to further improve the generalisability and accuracy of the method.

## 6. Conclusions

In this research, we explore the vulnerability of glacier debris flow along the G318 Linzhi section using remote sensing imagery and deep learning techniques. We have come to the following conclusions: (1) the precursors of glacier debris flows can be monitored and warned in real time using remote sensing technology; (2) glacier retreat and glacial lake formation are important factors in glacier debris flow susceptibility areas and should be given priority consideration; (3) with the help of remote sensing data, slopes, river valleys, cliffs, and water bodies can be effectively identified; and (4) by obtaining glacier morphological parameters, topographic slope and elevation data, and using deep learning techniques to construct complex predictive models, we can predict the susceptibility of glacier debris flows more accurately.

In the future, we will consider how to more accurately monitor changes in glacier edges, and use state-of-the-art deep learning methods to address monitoring changes in glacier edges, observing changes in glaciers, and providing data support for applications such as environmental protection.

## Figures and Tables

**Figure 1 sensors-23-06608-f001:**
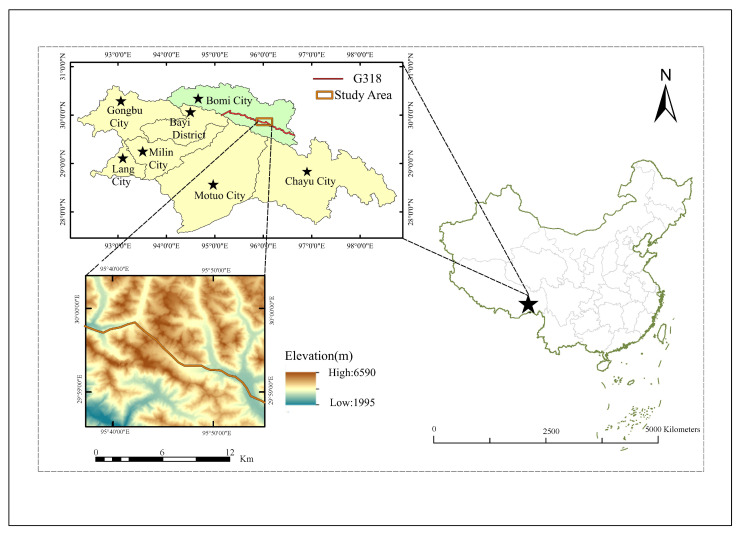
Geographical location of the glacier study area.

**Figure 2 sensors-23-06608-f002:**
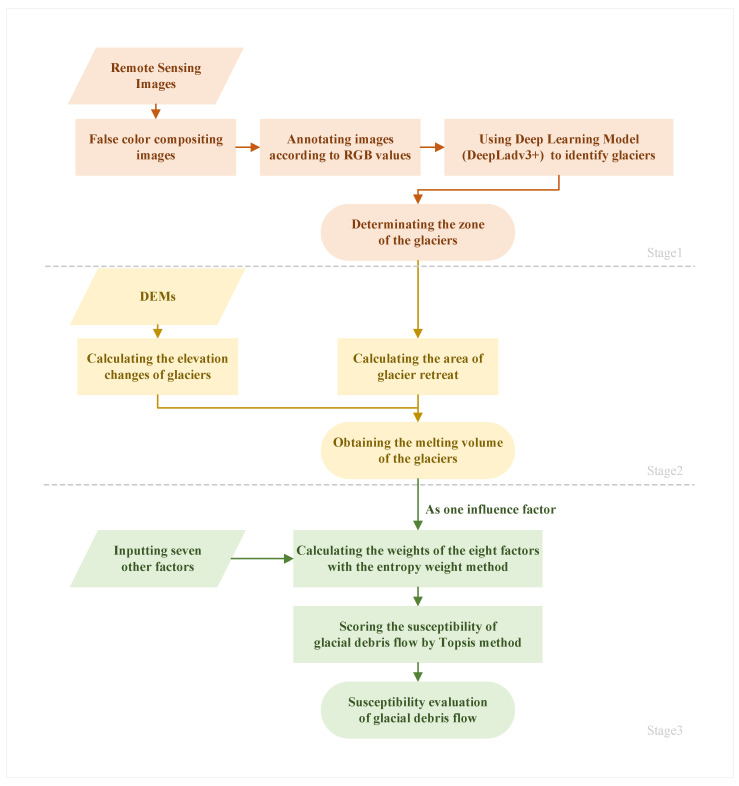
Workflow of the susceptibility analysis of glacier debris flow.

**Figure 3 sensors-23-06608-f003:**
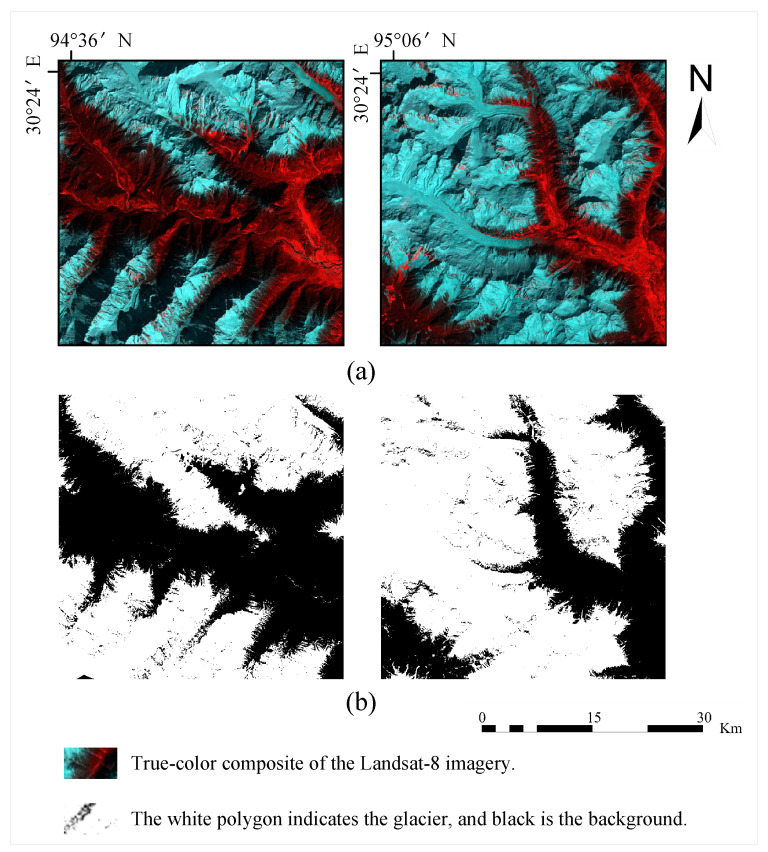
The dataset of semantic segmentation. Examples show the sample labels of glaciers in different images. (**a**) True-colour composite of the Landsat 8 imagery; (**b**) the white polygon indicates the glacier, and black is the background. (Band2 (Blue), Band3 (Green) and Band6 (SWIR 1)).

**Figure 4 sensors-23-06608-f004:**
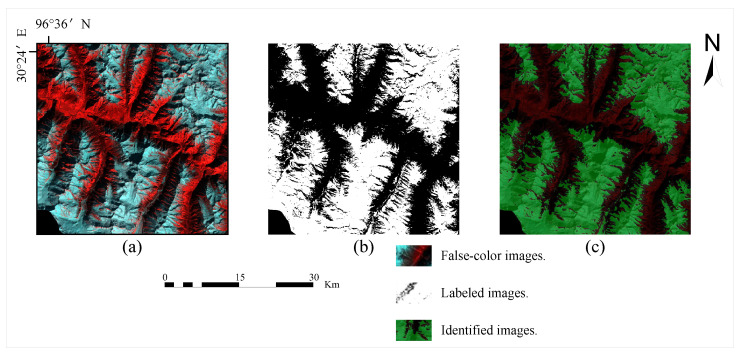
The processing of remote sensing images (**a**) False-colour images; (**b**) labelled images; (**c**) identified images.

**Figure 5 sensors-23-06608-f005:**
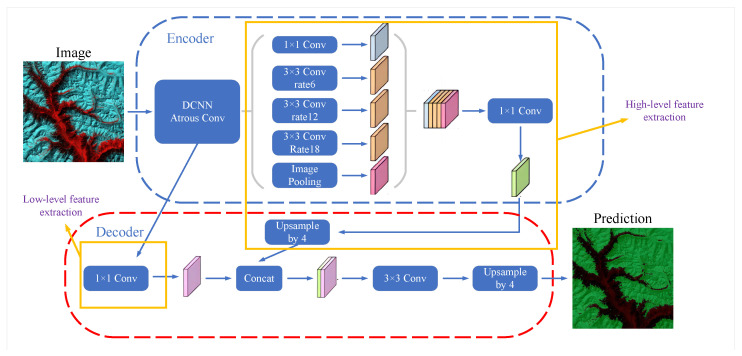
DeepLabv3+ architecture for semantic segmentation (figure adapted from [24]).

**Figure 6 sensors-23-06608-f006:**
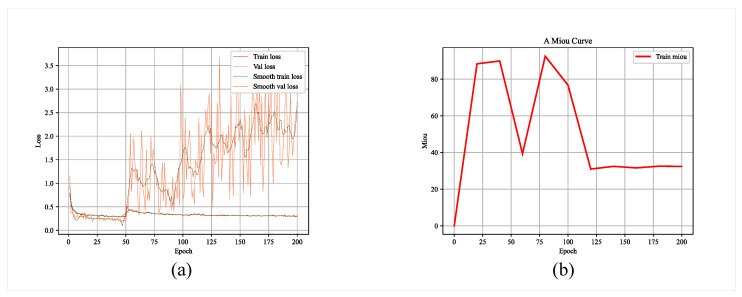
Model parameters after 200 epochs of training: (**a**) loss; (**b**) MIoU.

**Figure 7 sensors-23-06608-f007:**
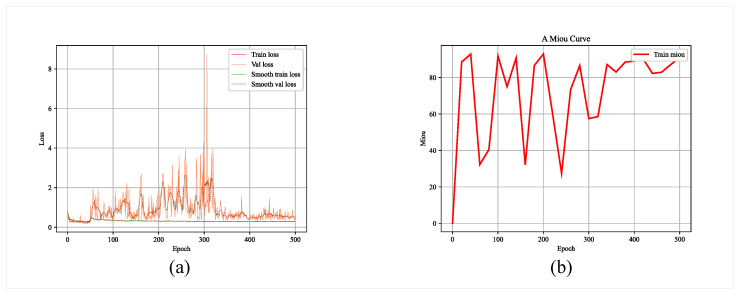
Model parameters after 500 epochs of training: (**a**) loss; (**b**) MIoU.

**Figure 8 sensors-23-06608-f008:**
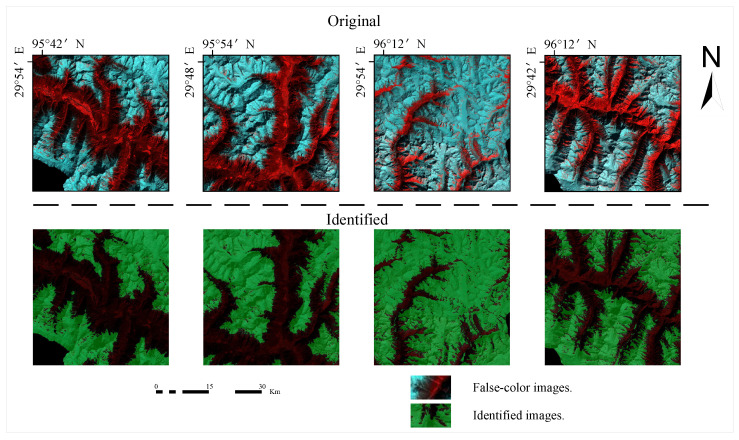
Comparison of the original image and the segmentation model results of glacier boundary part.

**Figure 9 sensors-23-06608-f009:**
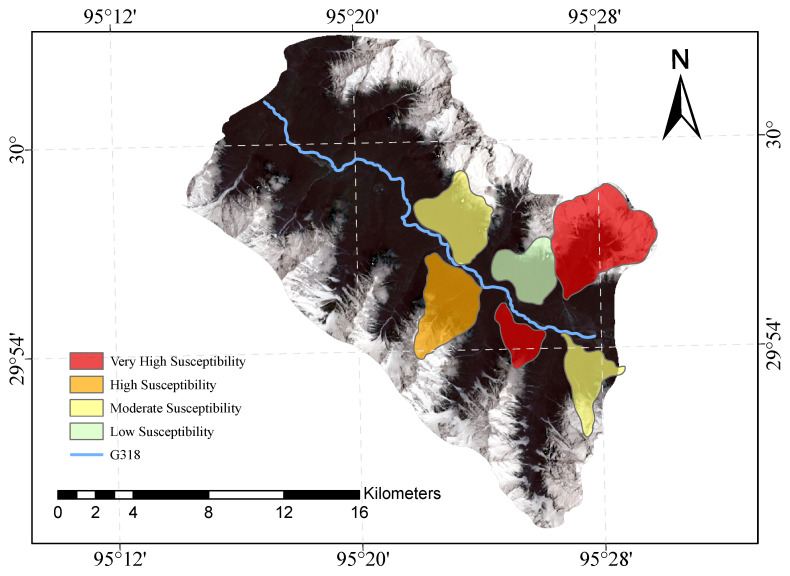
Glacial debris flow susceptibility mapping of the study area.

**Figure 10 sensors-23-06608-f010:**
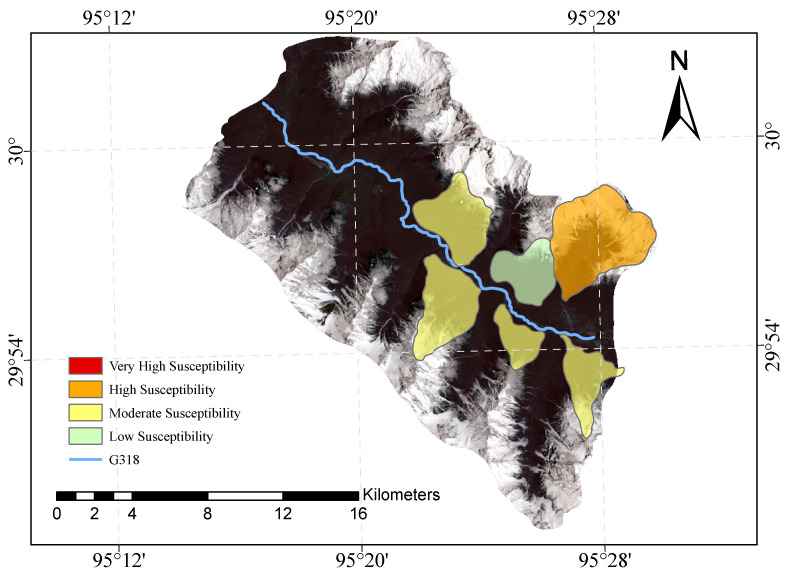
Previous glacial debris flow susceptibility mapping of the study area.

**Figure 11 sensors-23-06608-f011:**
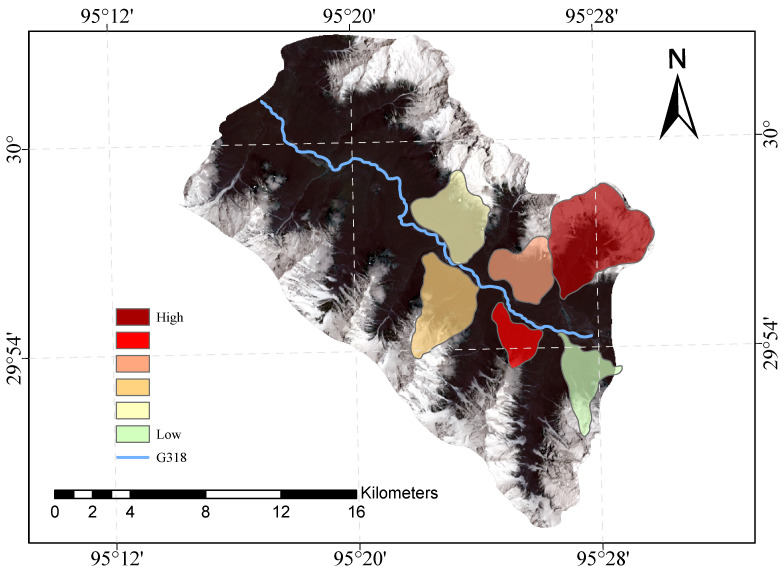
Comparison of the volume of the physical source in the study area.

**Figure 12 sensors-23-06608-f012:**
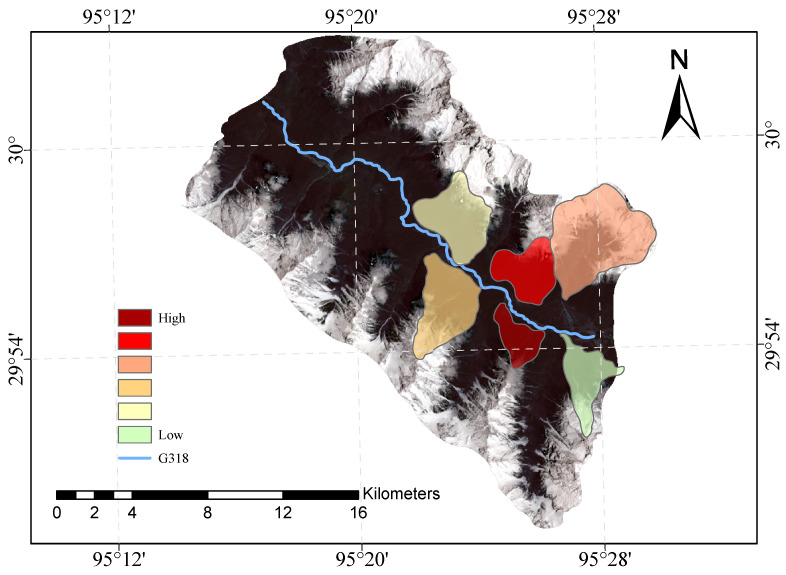
Comparison of catchment areas in the study area.

**Figure 13 sensors-23-06608-f013:**
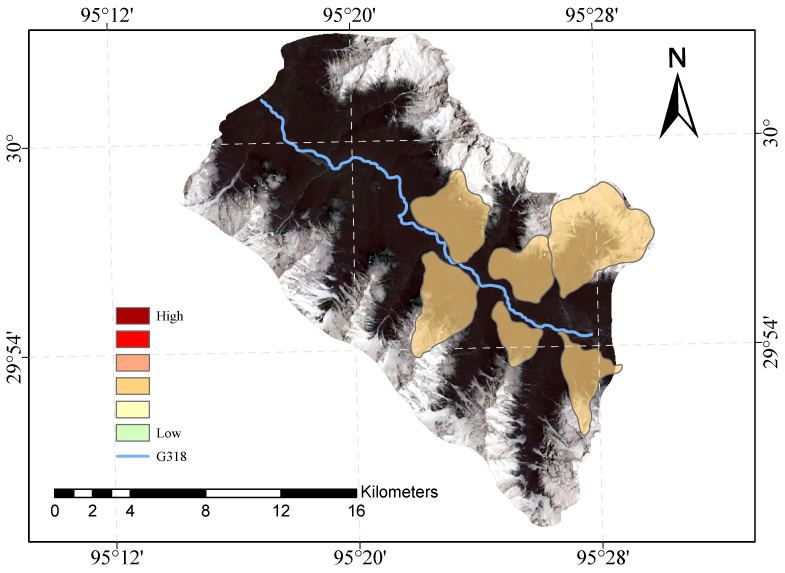
Comparison of maximum daily rainfall in the study area.

**Figure 14 sensors-23-06608-f014:**
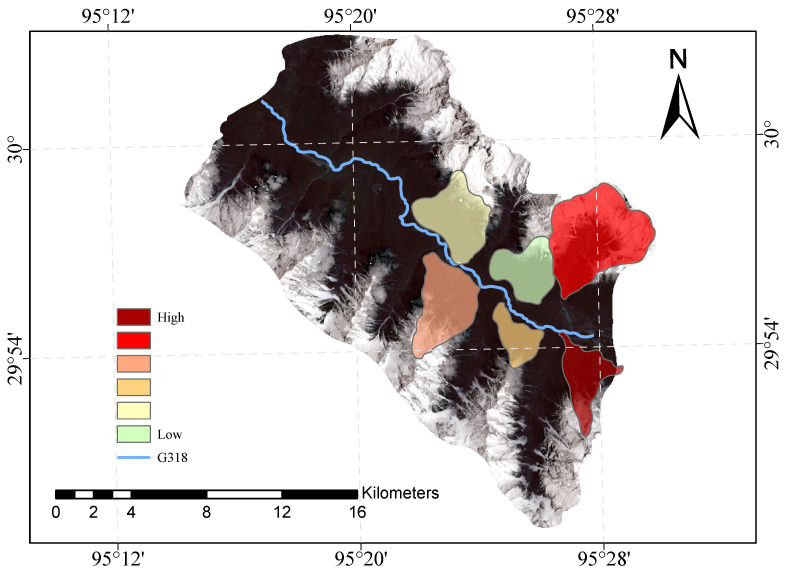
Comparison of the longitudinal slope drop of the main ditch in the study area.

**Figure 15 sensors-23-06608-f015:**
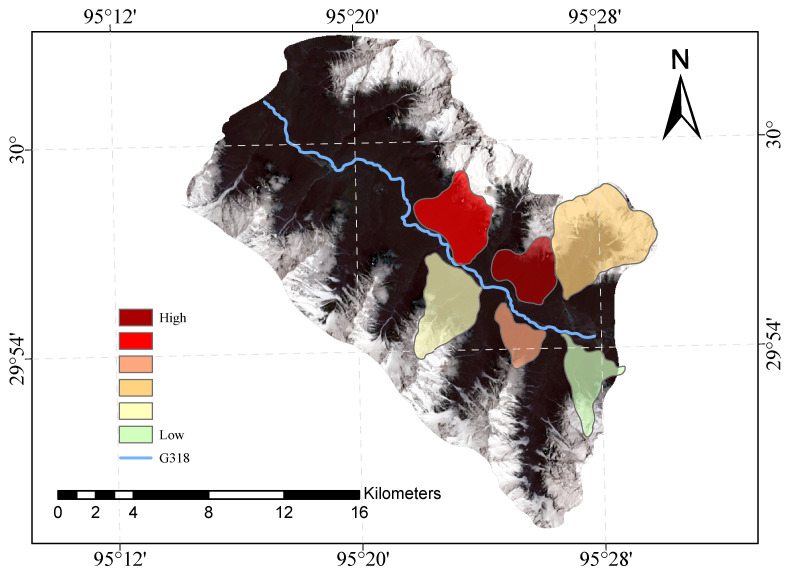
Comparison of the length of the main ditch in the study area.

**Figure 16 sensors-23-06608-f016:**
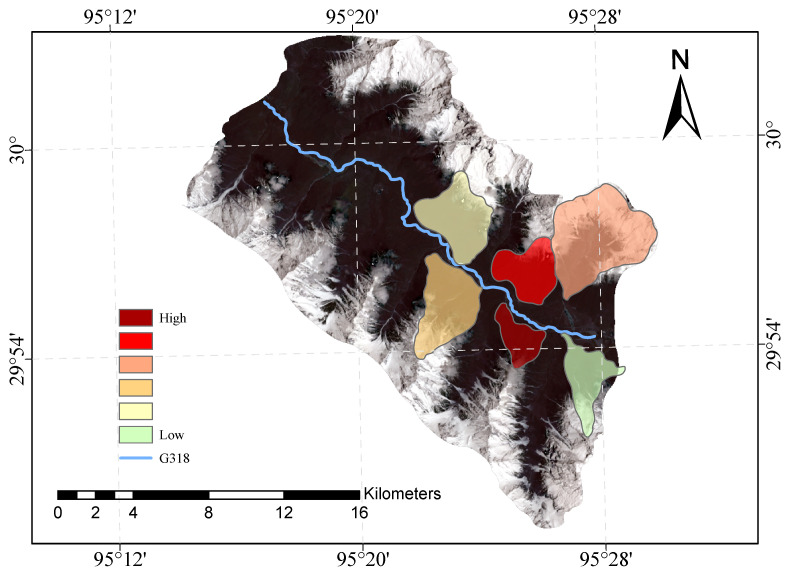
Comparison of glacier volumes in the study area.

**Figure 17 sensors-23-06608-f017:**
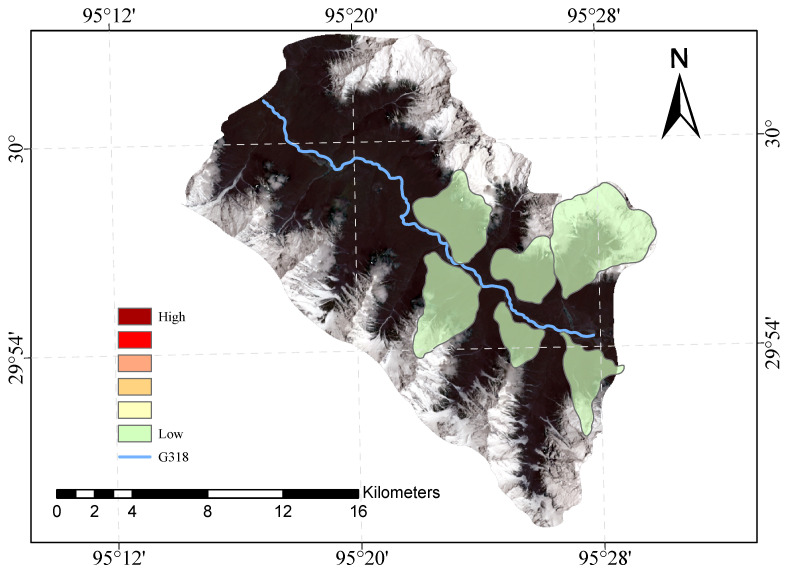
Comparison of the total glacial lake area in the study area.

**Figure 18 sensors-23-06608-f018:**
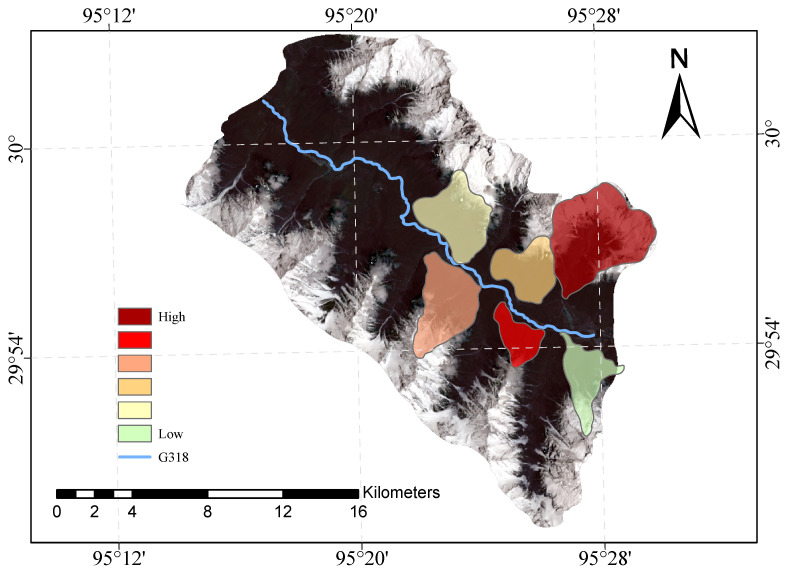
Comparative map of vegetation area in the study area.

**Table 1 sensors-23-06608-t001:** Evaluation metrics based on the entropy weighting method (EWM).

Influencing Factors	$X_{1}$	$X_{2}$	$X_{3}$	$X_{4}$	$X_{5}$	$X_{6}$	$X_{7}$	$X_{8}$
Weights	0.205	0.156	0.006	0.042	0.009	0.219	0.358	0.005

**Table 2 sensors-23-06608-t002:** Scores for six glacier debris flows based on the Topsis method of analysis.

No.	1	2	3	4	5	6
Score	0.060	0.071	0.055	0.097	0.083	0.048

## Data Availability

Not applicable.

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
