# Peer review of "Susceptibility Analysis of Glacier Debris Flow Based on Remote Sensing Imagery and Deep Learning: A Case Study along the G318 Linzhi Section"

_sensors, 2023, doi:10.3390/s23146608_

Round 1
Reviewer 1 Report
This paper presents the results of research „Susceptibility Analysis of Glacier Debris Flow Based on Remote Sensing Imagery and Deep Learning: A Case Study along the G318 Linzhi Section”.
The article is interesting. It analyzes an important problem with the risk of motorway mass movements. It shows the use of remote sensing and deep learning in the study of glacier debris flow.
However, the authors made many mistakes.
More specifically, I have the following main comments:
ü Section 1. Introduction – please remove -102-112.
ü Section 2. Study area- Fig.1 needs a complete change. in this form it isn’t interesting and, worst of all, it is not known where the research area is located.
ü Section 3. Methodology- Fig.2 - Please odd color to the individual stages.
ü Fig.3 - There are no explanations about the satellite images: What channels were used to show the Landsat images. The scale, north direction and geographic coordinates are missing.
ü Fig.4 – The same remarks as for Fig.3.
ü
ü 4. Section 5. Discussion – 5.1 - . What I miss in this section is a discussion of the advantages presented against the background of world literature. Based on your research results, please highlight the progress in the study of glacier debris flow. What has been studied so far?
ü
ü 5.2 - There is no critical discussion of the research results against the background of world literature
ü References
Please complete the missing references to the Discussion section.

Author Response
Dear Reviewer,
We would like to submit our research paper entitled “Susceptibility Analysis of Glacier Debris Flow Based on Remote Sensing Imagery and Deep Learning: A Case Study along the G318 Linzhi Section” for your consideration for publication in Sensors.
We have made a point-by-point response to the reviewers’ comments and suggestions, including a detailed description of any requested or suggested revisions.
We have also carefully checked and corrected the writing format and modified the structure and description of this paper to better fit the style of Sensors.
All the modifications and explanations in this revised version are listed in detail in the following “Responses to the Reviewers” and “Marked Manuscript”.
We would deeply appreciate your consideration and reviewers’ helpful comments and suggestions.
Yours Sincerely,
Jiaqing Chen, Hong Gao,Le Han, Ruilin Yu, Gang Mei*
School of Engineering and Technology,
China University of Geosciences (Beijing)
Email: gang.mei@cugb.edu.cn

Reviewer 2 Report
Dear Authors,
I have read your manuscript thoroughly, and I am pretty sure its content should somehow be published. The main idea refers to very vibrant issues in science; however, the current form of your text needs to be slightly improved.
General comment - before resubmission, please provide a thorough linguistic check - especially in terms of repetitions, style etc. I also spotted some typos, beginning sentences with small letters (e.g. line 56), and missing spaces (line 380), additional spaces (line 70), so please read the whole manuscript carefully and correct it where needed.
Moreover, please consider submitting your text to, e.g. "Remote Sensing" rather than "Sensors". In my opinion, your text definitely refers to remote sensing problems, and you didn't examine any particular sensor or measuring device.
Referring to the merit:
- Please justify choosing the exact test area (lines 114-120). It's true that the following lines 121-123 partially confirm that fact, but we can assume that there are many more similar landscape forms fulfilling your requirements.
- Each figure containing a map must be corrected regarding the scalebar! According to the cartographic principles, a scale must be divided into round, decimal values. No fractions are allowed!
- Figure 1 contains the wrong colour scale. It should be designed from the highest value (red, orange etc.) down to the lowest (green, blue). Currently, it looks the opposite. I also don't understand the role of the square field in the bottom right corner.
- All scales must be corrected in terms of their units. An internationally accepted norm is the SI, meaning that the miles must be changed into kilometres.
- Beginning from line 150: all positive temperatures must be assigned with a '+' sign.
- Line 185 - please complete the link with the accessibility date.
- Line 221: please justify choosing that particular neural network.
- Figure 5: if the chart is your finding, please express it more clearly; if it is not - please cite the relevant literature.
- Line 236: please justify choosing that particular classification model.
- Formulas 1,2,3 need an explanation of whether it is your finding or is it taken from literature.
- Line 291: please explain choosing that particular validating set and the exact number of pictures.
- Figure 6 is hard to read, especially regarding the chart lines. They look the same and overlap, which makes the whole perception difficult.
- The above comment also refers to Figure 7.
- Figures 9-18: each figure contains the wrong scalebar (please refer to my second comment about the maps) and - except for Figures 9 and 10 - wrongly designed colour scales (Fig. 9 and 10 are ok. - please correct the rest based on them).
- Lines 463-439 contain repetition that was already mentioned. The same was also repeated in lines 499-500.
- Lines 476-485 partially answer the question about choosing that particular classification model. However, this passage should be put before the considerations starting at point 3.4 (line 235). It means the text needs delicate restructuring.
Should the relevant corrections be introduced, the material needs to be resubmitted for another reviewing cycle. I wish you good luck!
The English language is generally understandable; however, the text contains repetitions and other errors needing improvements. I recommend checking the manuscript thoroughly before resubmitting it for another reviewing cycle.
Author Response

(The authors gave the same response as above.)

Reviewer 3 Report
Well written paper. I am not an expert in this area but found the paper very interesting and was able to follow the analysis.
Author Response

(The authors gave the same response as above.)

Round 2
Reviewer 1 Report
I haven't any comments.
Reviewer 2 Report
Dear Authors,
Thank you very much for your answers to my comments and suggestions. I read them carefully, and I entirely accept them. In my opinion, your article is much more transparent now and - what I find very important - its cartographic part definitely looks better. Hence, I do not have any other objections and accept your text for publishing. However, before proceeding, I suggest you recheck your text regarding potential errors.
I wish you good luck!